# Simultaneous Verification of Multi-Ultrasonic-Flowmeters in Tandem Arrangement for Natural Gas

**Tingyang Hong [1], Meng Yang [2], Chengxu Tu [1,3,4,*], Qipeng Huang [1], Fubing Bao [1,*], Zhaoqin Yin [1] and Xiaoyan Gao [1]**

1   Zhejiang Provincial Key Laboratory of Flow Measurement Technology, China Jiliang University, Hangzhou 310018, China
2   Petrochina West-East Natural Gas Pipeline Company, Nanjing 210058, China
3   College of Control Science and Engineering, Zhejiang University, Hangzhou 310027, China
4   LEO Group Co., Ltd., Wenling 317500, China
*   Correspondence: Tuchengxu@cjlu.edu.cn (C.T.); dingobao@cjlu.edu.cn (F.B.); Tel.: +86-571-87676277 (F.B.)



**Featured Application:** This study effectively guides the SVUFMST of Nanjing natural gas measurement station, which have greatly reduced the verification time and economic cost in practical work.

**Abstract:** A combined experiment and numerical analysis was performed to determine whether the flow field has a significant impact on the simultaneous verification of multi-ultrasonic flowmeters (UFMs) in tandem arrangement (SVUFMST) and whether the SVUFMST is practical for flowrate measurement of pressurized natural gas. In practical testing, SVUFMST can highly improve the efficiency of UFMs' calibration. Two types of UFMs of different nominal diameter were studied: DN250 and DN150. When it comes to DN250, data obtained from the actual-flow experiments were first used to verify the simulated results for $n \leq 3$ (number of tested UFMs) and then the verified numerical method was extended to simulate the internal flow of four and five tested UFMs. Comparison analysis shows that all the numerical results agree remarkably well with the available experimental data and more than two tested UFMs can induce an overall shift in the *E–Q* curve (relative measurement error *E* plotted as a function of flowrate *Q*). For the slight difference among the tested UFMs, the practicability of the SVUFMST at $n \leq 4$ (DN250) and $n \leq 3$ (DN150) are thereby proved, whereas the striking difference in *E* (~1.0%) between the most downstream UFM and the other UFMs for $n = 5$ (DN250), associated with the outstanding collapse onto the velocity profile along the acoustic paths, results in the invalidation of the SVUFMST.

**Keywords:** simultaneous verification of Multi-Ultrasonic-Flowmeters; natural gas; actual flow experiments; CFD

## 1. Introduction

Increasing attention is being paid to measurement technologies designed for natural gas owing to their clean and efficient characteristics with extensive use in energy supply [1–3]. Further, it has received attention for its extreme facility of transportation, widespread and vast reserves, ease of use, and lack of solid residual after burning. Natural gas currently makes up over 24% of the world's energy and 6.2% of China's energy. As one of the most important energy networks in China, the West–East Natural Gas Transmission Project (WENGTP) was established and has been service since 2001, transports over 72 billion cubic meters of natural gas from western China to eastern China

each year, and results in the alleviation of domestic energy tensions and improvement of the fuel structure [4,5]. An inevitable challenge, however, for natural gas transmission and trade is how to measure the natural gas exactly and reduce measurement uncertainty. Aiming at this problem, high-precision ultrasonic flowmeters (UFM) with different diameters are usually used to measure the natural gas flow in the pipelines because of their high accuracy and repeatability as well as their low flow resistance and disturbance. As a result, the verification of natural gas flowmeters is not only increasingly required but also, from the perspective of accurate energy consumption and settlement of work, is even more important. Unfortunately, the current verification method (JJG1030-2007 Verification Regulation of Ultrasonic Flowmeters [6]; ISO 17089-1:2010(en) Measurement of fluid flow in closed conduits [7]) is time-consuming and has low efficiency because the installation, disassembly, and verification of each flowmeter take one to two hours or more, and only one flowmeter would be tested in a calibration process according to the above standard regulations. Moreover, in order to accurately reproduce the operating environment, the UFMs in WENGTP would be calibrated in several important natural gas measurement stations along the mainline of WENGTP, such as Nanjing natural gas measurement station (NJMS) and Chengdu (CDMS) natural gas measurement station of China National Petroleum Corporation, and the natural gas flow in the mainline should be switched to the bypass line for UFMs' verification. Therefore, this metrology method would result in an unusual natural gas supply and urge a more efficient and practical verification regulation to reduce the disturbance of UFMs' verification on the natural gas supply. Consequently, increasingly more attention has been paid to the necessity of improving the efficiency of multi-UFMs' verification. As proposed and practically applied by some related manufacturers and metrological services, the simultaneous verification of multi-UFMs in tandem arrangement (SVUFMST) is a highly efficient metrology method to address this problem. However, there are only a few published studies regarding SVUFMST. Wan, who works in CDMS, calibrated two UFMs in tandem arrangement and analyzed the error values to confirm the flowmeter measurement performance, and his results showed that two flowmeters in a tandem arrangement can be simultaneously calibrated [8]. However, there is still no flow analysis or defined applicable conditions for SVUFMST. The schematic of the SVUFMST is illustrated in Figure 1, in which the tested UFMs were tandemly arranged downstream of the master meters and checking meters successively, with upstream and downstream straight pipes of appropriate length. Although this method has been applied in practical verification, it is still in the exploration stage and there is no corresponding operation standard accessible. Therefore, it is urgent to carry out comprehensive research on this method to perfect it.

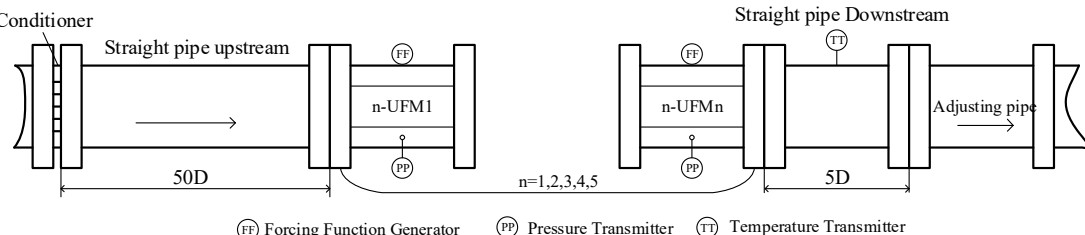

**Figure 1.** Schematic of the simultaneous verification of multi-ultrasonic flowmeters (UFMs) in tandem arrangement: FF: Forcing Function Generator; PP: Pressure Transmitter; TT: Temperature transmitter.

In this paper, the effect of the global flow field characteristics on the performance of the SVUFMST was studied to determine whether the flow field has a significant impact on the SVUFMST, and, if so, to explain the mechanism of its action. Although there are few studies that especially focus on the mechanism of the action of the flow field on SVUFMST, there are several trials concerning how the unusual flow fields can affect the UFMs measurement errors. Despite having low flow resistance and disturbance with extremely high precision, it is inevitable to absolutely exclude the local flow distortion, mainly in form of negative velocity, near the transducer recess (or protrusion)

in the ultrasonic flowmeter [9] in which an ultrasonic transducer is installed. Further, the negative velocities generated at the protrusion and recess locations as a whole are the key factors for negative measurement errors of the ultrasonic flowmeter. This result therefore confirms that the transducer recess in the ultrasonic flowmeter can affect the measurement accuracy. In addition to transducer recess and protrusion, the flow profiles [10,11], UFMs internal wall roughness [12], and pulsating flow [13] may all result in pronounced measurement errors of UFMs. It has been experimentally reported that the flow (velocity) profiles, such as the parabolic velocity distributions usually used for laminar and power law for turbulent flow, or a flattened airflow profile generated from a specially shaped convergent nozzle, also have a significant effect on the measurement error of ultrasonic meters [11], as does the complex flow profiles caused by the upstream elbow pipes as proposed by Zhao et al. [10]. Some of the above findings [9,10] suggest that computational fluid dynamics (CFD) can be an effective method to reveal the mechanism of action of the flow field on UFMs' measurement accuracy, and thus, in this study, CFD is adopted to numerically simulate the internal flow field in the tandem multi-UFMs and help quantify the influence of the flow field on the SVUFMST performance.

Further, in terms of SVUFMST, the influence of the detailed field characteristics of the flow on the ultrasonic flowrate signals, such as the upstream flow distortion near the transducer recess affecting the downstream flowmeter measurement, cannot be determined only from experimental results, because there always exists uncertainty resulting from the product differences in large-quantity production. Compared with experimental methods, however, the simulation results using CFD can convincingly exclude the disturbance of industry manufactures and are highly focused on the aerodynamic factor. By comparing the preliminary simulation results to the available experimental data, the grids, boundary conditions, and key model parameters were carefully adjusted to meet the numerical accuracy and maximum permissible error of UFMs, and then the optimized numerical model was extended to simulate the internal flow of UFMs in tandem arrangement in other severe conditions. This probably leads to the inability or excessive cost of executing the experiments for three or more UFMs tested in tandem arrangement or different types of UFMs in SVUFMST. Our study provides some effective guidance for the SVUFMST of Nanjing natural gas measurement station, which have greatly reduced the verification time and economic cost in practical work.

## 2. Materials and Methods

### 2.1. Experimental Method

The SVUFMST is depicted schematically in Figure 1 and the tested UFMs arranged in tandem along the streamwise direction were denoted as UFM1 to UFMn (here, $n \leq 5$) at different testing locations. $n$ is the number of tested UFMs, which are prefixed to indicate the total number of tested UFMs, for example, 5-UFM1. In the present study, a type of transit-time UFMs (Daniel 3400) were examined, as shown in Table 1, using the working standard facility (WSF) in NJMS, and according to the China national verification regulation of UFMs "JJG 1030-2007" and ISO 17089-1:2010(en), "Measurement of fluid flow in closed conduits-Ultrasonic meters for gas—Part 1: Meters for custody transfer and allocation measurement." These target flowmeters, with two nominal diameters, DN250 and DN150, have two sets of four paths, namely, two cross planes for the four paths, as shown in Figure 2. The abscissa, as presented in Table 1, means the plane of Path A and D is 0.809R away from the horizontal center plane of pipeline and Path B and C is 0.309R close to the center plane of pipeline. The expanded uncertainty (k = 2) of the WSF is within 0.29% of the flowrate range of 8–12,000 $m^3/h$, which is used to verify the DN50–DN400 flowmeters with their accuracy no higher than 1.0%. Furthermore, the experimental data measured from NJMS and the experimental environment provided has strong traceability, with the help of the established primary standard facility (the expanded uncertainty is 0.10% and k = 2) and secondary standard facility (the expanded uncertainty is 0.25% and k = 2).

**Table 1.** Technique parameters of the tested ultrasonic flowmeter.

| Model | Path Type | Size | Abscissa | Inclination Angle of Acoustic Paths | Flowrate (m³/h) | Accuracy (%) |
|---|---|---|---|---|---|---|
| Daniel 3400 | Four paths | DN250 | 0.309R (Path B and C) 0.809R (Path A and D) | 60° | 153–5084.96 | 1.0 |

R is the radius of the tested UFM (Daniel 3400).

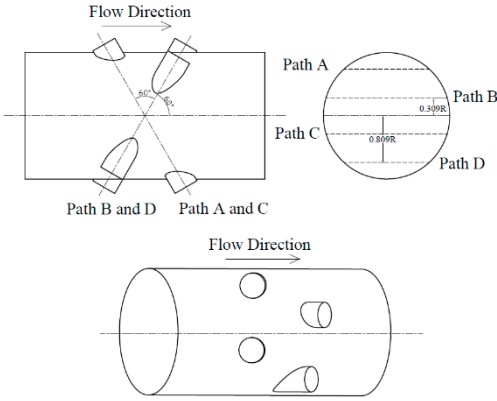

**Figure 2.** Schematic of the acoustic path arrangement in the tested UFM (Daniel 3400): the upper right corner is the side view of the upper left corner (the recesses are hidden).

The layout of the WSF is illustrated in Figure 3, which is designed as a bypass conduit of the mainline of WENGTP to achieve an actual-flow calibration. Once a UFMs verification is being conducted, the natural gas flow in the mainline would be switched preliminarily to the bypass conduit (also known as WSF). By applying the "master meter method" [6,7], the facility consists of three flowrate-measurement nodes: the master meters (MM), checking meters (CM), and UFMs under test; the flow is generated by the pressure drop between the upstream inlet and downstream outlet of the WSF, as a result of the driving force of WENGTPs mainline. The checking meter is used to monitor the performance of the master meter in real time and the flowrate tested by the master flowmeters are designated as the reference flowrate. In order to improve the full-scale performance of WSF, this verification system is divided into two sub-systems aligned in parallel, depending on the flowrate scales: large working standard for the high-flowrate verifications and small working standard for the low-flowrate verifications, as shown in Figure 3. Therefore, the flowmeters and straight pipes with larger inner diameters are employed in the large working standard. As for the large working standard, the master meter is composed of 11 turbine flowmeters, three of which are DN150 and eight of which are DN200. Then, the 11 corresponding pipelines of the master meter converged into two checking meter lines equipped with two UFMs (DN150 and DN400), either of which could be selected to connect with the tested flowmeter of the inner diameter ranging from DN150 to DN400. Compared with the large WSF, three smaller turbine flow meters (DN50, DN80, and DN100) as the master meters and a single DN100 UFM as the checking meter were utilized to verify the tested flowmeters with small inner diameters (DN50, DN80, and DN100) in the small WSF. FE20 and FE16, as depicted in Figure 3, represent the UFMs and the turbine flowmeters, respectively, and flow control valves are installed both upstream and downstream of each flowmeter in WSF. The straight pipe length upstream and downstream of each flowmeter under test was maintained higher than 150D and 20D, respectively, to achieve a fully developed inner flow. Figure 1 also shows the straight pipe layout and Table 2 lists the straight pipe length for different flowmeters. As a mixed-flow medium (composed of approximately 95% methane, 3% ethane, 1% nitrogen, 0.5% carbon dioxide, and partial alkanes), the temperature, pressure, and chemical composition of the tested natural gas had been simultaneously measured to obtain its actual density and viscosity in each experiment. Some of its physical and

chemical properties are listed in Table 3. Here, the chemical composition analysis was performed using a gas chromatograph and pressure gauges and thermometers were installed as shown in Figure 1.

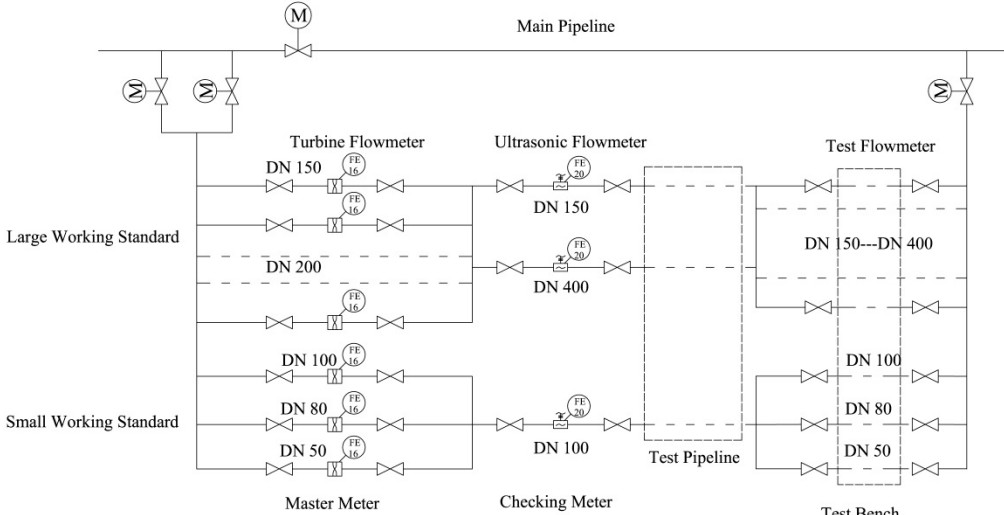

**Figure 3.** Schematic of the working standard facility for verification of the tested UFMs.

**Table 2.** Straight pipe configuration in the working standard facility.

|  | Straight Length Upstream | Straight Length Downstream |
|---|---|---|
| **Master Meter** | 25D | 5D |
| **Checking Meter** | 25D | 5D |
| **Test Meter** | 150D | 20D |

**Table 3.** Physical and chemical properties of the pressurized natural gas under test.

| DN250 | 1-UFM1 | 2-UFM1 | 2-UFM2 | 3-UFM1 | 3-UFM2 | 3-UFM3 |
|---|---|---|---|---|---|---|
| **$CH_4$ (wt.%)** | 94.7567 | 94.7637 | 94.7546 | 94.7589 | 94.7546 | 94.7560 |
| **$N_2$ (wt.%)** | 1.2633 | 1.2604 | 1.2596 | 1.2600 | 1.2596 | 1.2609 |
| **$CO_2$ (wt.%)** | 0.7658 | 0.7493 | 0.7599 | 0.7533 | 0.7599 | 0.7644 |
| **$C_2H_6$ (wt.%)** | 2.8918 | 2.9065 | 2.9039 | 2.9084 | 2.9039 | 2.8970 |
| **$C_3H_8$ (wt.%)** | 0.1964 | 0.1949 | 0.1960 | 0.1943 | 0.1960 | 0.1963 |
| **P (MPa)** | 6.5 | 6.5 | 6.5 | 6.5 | 6.5 | 6.5 |
| **T (K)** | 293 | 293 | 293 | 293 | 293 | 293 |
| **$\rho$ (kg/m$^3$)** | 51.278 | 51.268 | 51.277 | 51.272 | 51.277 | 51.279 |
| **$\mu$ (Pa·s)** | 1.2583 | 1.2582 | 1.2583 | 1.2583 | 1.2583 | 1.2583 |

Usually, regardless of whether a single UFM or multi-UFM is verified, each flowmeter is tested at a five standard flowrate ($Q_{min}$, $Q_t$, $0.4Q_{max}$, $0.7Q_{max}$, and $Q_{max}$) required by the national standard; the test is repeated three times at a certain flowrate. Each target flowmeter had been verified alone using the WSF before implementing SVUFMST to exclude the incorrect UFMs. It should be noted that as SVUFMST is carried out, any unusual relative difference, such as one beyond the maximum permissible error of the tested UFMs, between the flowrate tested by a single meter and that by the tested multi-UFMs would invalidate the SVUFMST, despite the number of tested UFMs, hereinafter denoted as *n*.

### 2.2. Numerical Methods

To explore the effect of the flow disturbance caused by the transducer recesses on the performance of the simultaneous verification of multi-UFMs arranged in tandem, we consider five cases with different numbers of tested UFMs ranging from 1 to 5 ($1 \leq n \leq 5$), the geometries and configurations of

which were carefully measured and modeled according to the actual UFMs. In addition, in order to deepen the understanding of the influence of the change of flow field on SVUFMST, we also consider three cases with same type but different diameters (DN150). Figure 2 depicts a wetted UFM Daniel 3400 (DN250 and 150) studied in this paper, which has four paths and eight transducer recesses along the paths in different configurations. The simulations were carried out in natural gas using 50D and 10D straight pipes upstream and downstream of the tested UFMs, respectively. The natural gas is set as an incompressible mixed-flow medium according to the experimental conditions. The 50D straight pipe upstream was determined by conducting numerical verifications based on the previous work supposed by Cimbala et al. [14] to achieve the fully developed flow at the 1st tested UFM inlet. The constant axial flow velocity $U_0$ (for DN250) ranged from 0.88 m/s to 29.22 m/s and the fully developed outflow were used as the inlet and outlet boundary conditions, respectively, in our CFD simulations. In this study, the relatively high pressure in the pipe and flowrate led to a very large Re in the range of 106–107 and, hence, only turbulent flow needs to be considered.

In consideration of the high accuracy of the tested UFMs (1.0% full scale), it is very important to obtain reasonable calculation accuracy for examining the SVUFMST's performance. Thus, we use 3D CAD software to build the model according to the actual geometric measurement results of the tested flowmeter. Then, we use advanced polyhedral mesh technology to render the mesh and refine it in both near-transducer-recesses regions and flowmeter bodies. Because considering the high Reynolds number environment in which the inertial force dominates the flow in the tube while the viscous force affects the near-wall and near-transducer-recesses flow, we also add boundary layer grids in these regions with a height of 0.0001D of the first layer of elements and an exponential growth row (growth factor: 1.2, rows: 10). Figure 4 depicts the grid of a single UFM and a detailed profile of the recesses. This mesh improves the prediction accuracy of the velocity profiles, especially the steep velocity gradients on the near-wall region and negative velocity near transducer-recesses region. Here, the velocity profile is the definition given for the distribution of velocities in the axial direction over the cross section of the circular pipe or along the acoustic paths of a UFM. For different numbers of tested UFMs, the total cell number was in the range of 5–15 million after verifying the grid-independence. The finite volume solver in Fluent was used to calculate the time-averaged Navier–Stokes equations and the k-epsilon closure equations. The realizable k-epsilon turbulence model in combination with the wall treatment of SWF was finally employed in our simulations because it provided the best agreement with the experimental data obtained from a single tested UFM and better prediction for flow separation into the recesses after comparison with other commonly used turbulence models. Furthermore, the second-order upwind discretization scheme and SIMPLE algorithm were adopted.

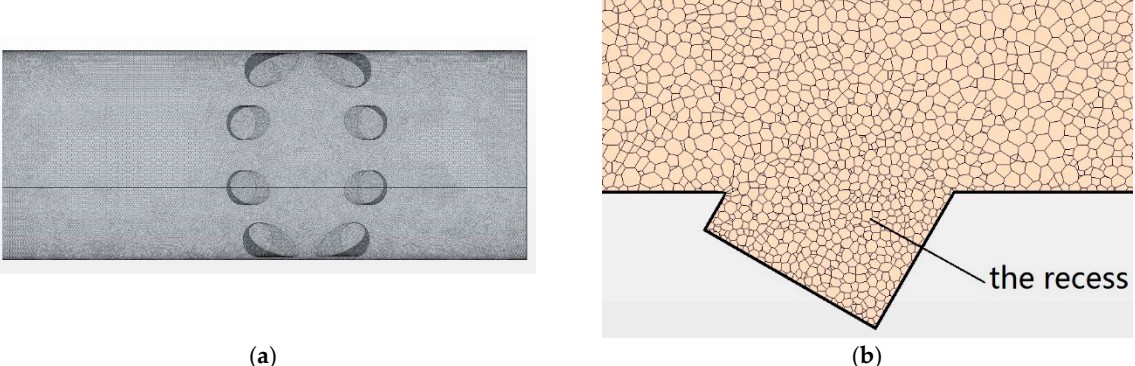

(**a**)    (**b**)

**Figure 4.** Schematic of the grid: (**a**) a single UFM; (**b**) detail profile of the recess.

According to the manufacturer's instructions (Emerson Daniel-3400 DN250 and 150), the mean axial velocity measured by the UFMs can be calculated from

$$U_{\text{UFM}} = w_{\text{A}} \times U_{\text{A}} + w_{\text{B}} \times U_{\text{B}} + w_{\text{C}} \times U_{\text{C}} + w_{\text{D}} \times U_{\text{D}},\tag{1}$$

where $w$ is the weight associated with chordal velocity, subscripts A to D indicate the path A to D, respectively, and thus $U_{\text{A}}$ is the velocity measured along path A, etc. As for Daniel 3400, the weight coefficients are 0.1382 along paths A and D ($w_{\text{A}} = w_{\text{D}} = 0.1382$) and 0.3618 along paths B and C ($w_{\text{B}} = w_{\text{C}} = 0.3618$). Similarly, the space-averaged numerical velocity on a path was first obtained by extracting the velocity profile on that path from the numerical velocity field, which was then used to calculate the numerical mean axial flow velocity with the help of the same weight coefficients mentioned above. For both the experimental measurement and numerical simulation, the relative measurement error $E$ can be expressed as

$$E = \frac{Q - Q_{\text{in}}}{Q_{\text{in}}} \times 100\%,\tag{2}$$

where $Q$ can be calculated either from the UFM measured or from the UFM attained by post-processing the simulated results, and the corresponding $Q_{\text{in}}$ is either the standard flowrate generated by a flow-standard facility or the inlet flowrate given in advance as a boundary condition.

## 3. Results

### 3.1. Numerical Simulation Verification by Experiments for 1–3 Tested DN 250 UFMs

For one, two, and three UFMs in tandem arrangement, the relative measurement errors $E$ are plotted in Figure 5a–c as a function of flowrate $Q$, respectively. Both the experimental data (distributed symbols) and numerical results (continuous lines), can be fully described by a negative power-law relationship with two assigned constant coefficients $A$ and $B$, which can be expressed as

$$\ln(E + k) = A \ln(Q) + B,\tag{3}$$

The values of $A$ and $B$ for different numbers of UFMs and goodness of fit $R^2$ are listed in Table 4. An $R^2$ value very close to 1 indicates high goodness of fit of the simulated results. In addition, we add a standardized residual $\delta^*$ to observe the regression fit degree of the experimental value and the simulation curve. Once $\delta^*$ is in the range of $(-2, 2)$, we think the fitting is almost satisfactory.

**Table 4.** Constants in the fitting functions modeling the negative power-law *E–Q* curves.

| *n*-UFM*n* | *A* | *B* | *k* (Intercept) | Numerical R$^2$ | Experimental Standardized Residual δ* |
|---|---|---|---|---|---|
| 1-UFM1 | −0.427 | 2.037 | 0 | 0.967 | (−1.150~0.732) |
| 2-UFM1 | −0.432 | 2.219 | 0 | 0.962 | (−0.781~0.813) |
| 2-UFM2 | −0.403 | 2.040 | 0 | 0.959 | (−1.531~1.138) |
| 3-UFM1 | −0.325 | 1.416 | 0 | 0.996 | (−0.356~0.342) |
| 3-UFM2 | −0.261 | 1.166 | 0 | 0.951 | (−0.907~1.661) |
| 3-UFM3 | −0.288 | 1.393 | 0 | 0.994 | (−0.636~1.180) |
| 4-UFM1 | −0.307 | 1.584 | 0 | 0.978 | |
| 4-UFM2 | −0.349 | 1.880 | 0 | 0.934 | |
| 4-UFM3 | −0.558 | 3.037 | 0 | 0.853 | |
| 4-UFM4 | −0.520 | 2.680 | 0 | 0.910 | |
| 5-UFM1 | −0.277 | 1.428 | 0 | 0.978 | |
| 5-UFM2 | −0.305 | 1.602 | 0 | 0.968 | |
| 5-UFM3 | −0.367 | 1.984 | 0 | 0.929 | |
| 5-UFM4 | −0.408 | 2.210 | 0 | 0.918 | |
| 5-UFM5 | −0.309 | 1.599 | 1.713 | 0.996 | |

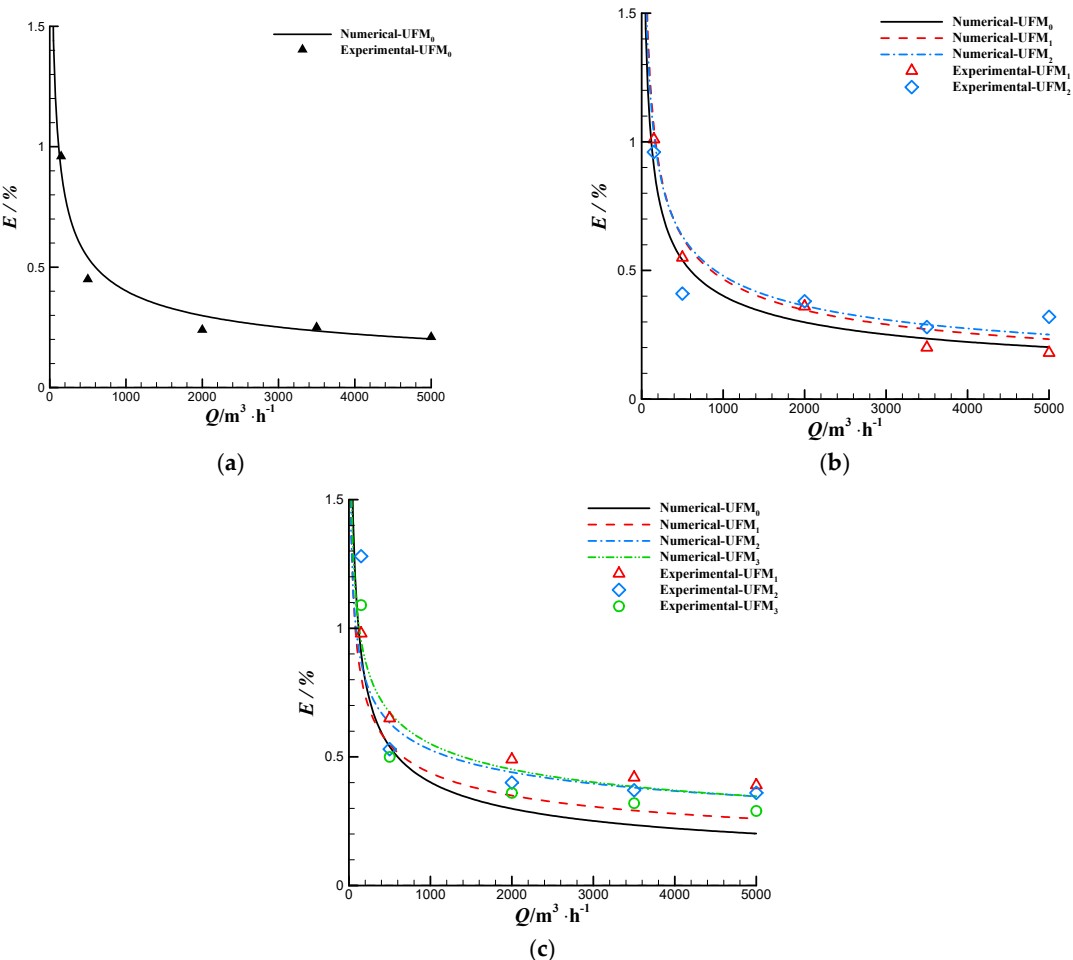

**Figure 5.** Relative measurement error $E$ as a function of flowrate $Q$ for 1–3 tested flowmeters: (**a**) a single tested UFM; (**b**) two tested UFMs; (**c**) three tested UFMs.

It is clear that $E$ decreases nonlinearly and monotonically with increasing $Q$ and the attenuation ratios for $Q < 600$ m$^3$/h are significantly larger than those for $Q > 2000$ m$^3$/h. Similarly, the approximately negative power-law relationship between $K_S$ and $Q$ can be found in Figure 5 and this phenomenon shows that the swirl secondary flow would be reduced effectively as $Q$ (or Re) increased. The above discussions indicate that all the available experimental data agree remarkably well with the numerical results and especially the overall $E$–$Q$ power-law distribution for a single tested UFM is in good accordance with the previous results [12,15,16]. Because the volumetric flowrate measured using UFMs is calculated as the product of the mean axial flow velocity $U_0$ and the area $S$ over the cross section, the velocity distribution correction factor $C$ (also known as velocity profile factor), which is used to evaluate $U_0$ corresponding to the uniform velocity profile (fully developed and usually turbulent), must be known [15]. The negative power-law relationship between $E$ and $Q$ for $1 \le n \le 3$ in geometric tandem arrangement can be mainly attributed to the change in $C$ as a function of Reynolds number Re, which can be calculated approximately for the fully developed velocity distribution in the axisymmetric non-swirling flow. Re can be calculated using Re = $UD/\nu$, where $D$ is the pipe or UFMs internal diameter and $\nu$ is the kinematic viscosity of the fluid. For a constant internal diameter of UFM and fluid viscosity, Re should increase linearly with increasing flowrate. Therefore, once the decrease in $Q$ decreases Re to a certain extent, the transition from a flattened velocity profile to a more convex velocity profile may occur, resulting in the evolution of the velocity distribution and the variance in $C$. Here, we only took $Q$ (equivalent to Re) and $n$ into consideration for our simulations, and other factors would be analyzed according to the difference $\delta$ between the experimental and simulated E, whereas

previous studies had suggested that $E$ depends on Re, pipe roughness, inlet flow conditions, distance from an inlet, etc. [12,15–17]. $\delta$ can be calculated by

$$\delta = E_t - E_s, \tag{4}$$

where the subscript t indicates the test results and s indicates the simulated results.

The maximum deviation $\delta$max between the experimental and simulated $E$ is about 0.06% for a single UFM at $Q = 500\ \mathrm{m^3/h}$ and $2000\ \mathrm{m^3/h}$, 0.18% for UFM2 of the two tested UFMs at $Q = 500\ \mathrm{m^3/h}$, and 0.18% for UFM3 of the three tested UFMs at $Q = 500\ \mathrm{m^3/h}$. Compared with the maximum permissible error of the UFM (Daniel-3400) of 1.0%, the value of $\delta$ as discussed above can be acceptable, and also confirms the feasibility of implementing CFD simulations in UFMs' internal flow and then calculating the measurement errors. It was clear that $\delta$max always occurred in the flow transition regime with $Q$ approximately ranging from $500\ \mathrm{m^3/h}$ to $1000\ \mathrm{m^3/h}$. As illustrated in Figure 5, the deviation $\delta_{\mathrm{UFM}}$ between the upstream and downstream UFMs' $E$ (adjacent or nonadjacent) is smaller than 0.15% with the only one exception ($\delta_{\mathrm{UFM}} = 0.3\%$) that can be found at $Q = Q_{\min}$ for three tested UFMs. It is noted that $\delta_{\mathrm{UFM}}$ is different from $\delta$, and is given as follows:

$$\delta_{\mathrm{UMF}} = E_{i+j} - E_i \ (i = 1, 2, 3, \ldots, n-1; \ j = 1, 2, 3, \ldots, n-i), \tag{5}$$

where a larger subscript number indicates a greater distance from the flow inlet at the test bench and $n$ is the number of the tested UFMs. Overall, the simulated $|\delta_{\mathrm{UFM}}|$ was smaller than the experimental $|\delta_{\mathrm{UFM}}|$ because only the flow field effect on $E$ was taken into consideration for the simulated results, whereas there were other factors introduced previously, such as UFMs' manufacturer and pipe roughness, affecting experimental data and then affecting $\delta_{\mathrm{UFM}}$. Thus, it can be observed that the magnitude of $\delta_{\mathrm{UFM}}$ is similar to that of the standard facilities' uncertainty, which is equivalent to 0.29% (k = 2) and can be regarded as an important inlet flow condition. Therefore, the uncertainty is large enough to change $\delta_{\mathrm{UFM}}$ from negative to positive values. In other words, it is ambiguous whether $E$ of the upstream UFM is larger than $E$ of the downstream UFM, as shown in Figure 5. From two tested UFMs to three tested UFMs, the absolute values of the measured $\delta_{\mathrm{UFM}}$ hardly changed, whereas those calculated from simulated flow fields increased obviously with increases in $n$. For three tested UFMs, the simulated $E$–$Q$ curves were concentrated in two nonlinear striped regions and the $E$–$Q$ curve of UFM1, which exclusively occupied the lower striped region, showed a clear shift to a smaller relative error, while $E$ of both downstream UFM2 and UFM3 were very close to each other. The two concentrated striped regions hereinafter can also be found in flow through more than three tested UFMs. It is of interest to note that the increase in $n$ can induce an overall shift in the $E$–$Q$ curve and the two concentrated striped regions are consequently built on the abrupt shift in an $E$–$Q$ curve associated with $n$.

### 3.2. Numerical Simulation Verification by Experiments for 4–5 Tested DN 250 UFMs

For four and five tested UFMs, only the numerical results were further presented and analyzed, owing to the lack of an adequate number of the same-type tested UFMs (Daniel 3400) in NJMS and the severe work schedule disturbance resulting from highly time-consuming experiments in this study, especially mounting and remounting the tested UFMs. Even so, the numerical methods used for the less-tested UFMs were verified in the previous section.

Similar to the case of three tested UFMs, the $E$–$Q$ curves for four tested UFMs were also concentrated into two distinctly separate striped regions and exhibited a negative power-law shape, but the $E$–$Q$ curves of UFM3 and UFM4 were closer to that for a single UFM in the lower striped region, whereas those of UFM1 and UFM2 were distributed in the upper striped region, as depicted in Figure 5. Compared with the UFMs' maximum permissible error of 1.0%, the average deviation between the lower and the upper striped region are roughly 0.15%, as $Q > 1000\ \mathrm{m^3/h}$, and then could be acceptable. This also suggests that it is practical to simultaneously verify four tested UFMs in a tandem arrangement. Furthermore, the increase in $n$ from three to four may be a trigger for the transposition

between $E$–$Q$ curves of downstream UFMs (UFM3, UFM4, or UFM5) and those of upstream UFMs (UFM1 or UFM2), and a similar behavior was found for five tested UFMs. The flow through from UFM2 to UFM3 led to the appearance of the two power-law striped regions of $E$–$Q$ curves for four tested UFMs, namely, a pronounced shift in the $E$–$Q$ curves, whereas that from UFM4 to UFM5 does so for five tested UFMs. These results imply that $n$ probably decreases the similarity of flow fields at different testing locations, especially the velocity profiles on the same-type paths, which may be partly attributed to the flow distortion at the transducer recesses due to the relatively clear negative velocity and difference in U-Profiles along paths. Thus, it is reasonable to presume that SVUFMST would perform more poorly with more tested UFMs.

It is noted that once an $E$ difference is obtained beyond 1/3 of the maximum permissible error, $E_{\max}$, of the tested UFM (Daniel 3400), SVUFMST cannot be conducted according to the verification regulation of UFMs (JJG1030-2007). For five tested UFMs, there were still two power-law striped regions into which the $E$–$Q$ curves of UFM1-UFM4 and UFM5 are concentrated, respectively. Unlike the grouping of the former $E$–$Q$ curves, only the $E$–$Q$ curve of UFM5 concentrated into the lower striped region, which was, as in our study, the only negative striped region; the other four $E$–$Q$ curves were uniformly distributed, but the UFM located more upstream was associated with the larger relative error E, as mentioned above. The striking average difference of $E$ between the upper striped region and the lower striped region is higher than 1% on the same level as $E_{\max}$, as shown in Figure 6b. Table 4 lists the important parameters for the best-fit power-law expressions, in which the sudden decrease in the intercept $k$ of the $E$–$Q$ curve at UFM5 is equal to 1.713, whereas $k$ of the other $E$–$Q$ curves ($n = 5$) is zero. This suggests that as the gas flows through five tested UFMs, the flow distortion was significantly strengthened, and the decrease in the internal flow uniformity significantly destroyed the consistency of the UFMs' measurement conditions. Therefore, the invalidation of the SVUFMST for $n = 5$ is numerically proved.

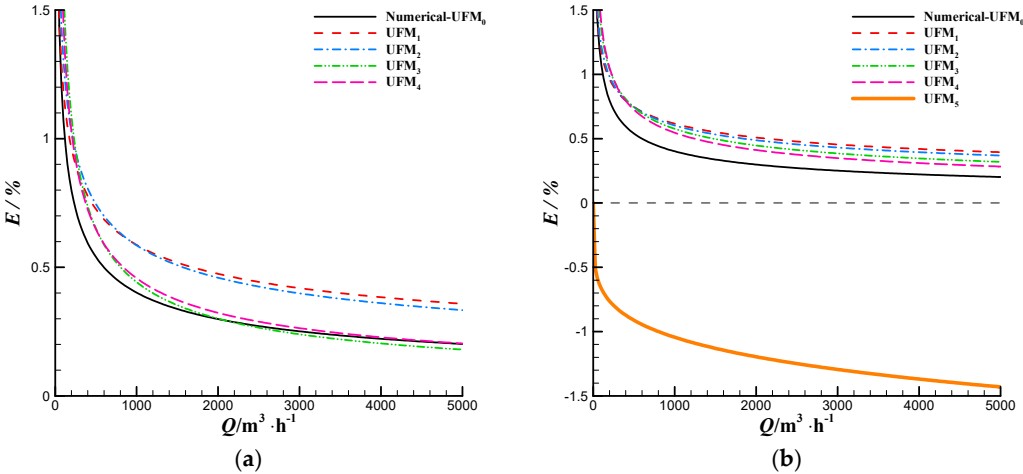

**Figure 6.** Relative measurement error $E$ as a function of flowrate $Q$ for four to five tested flowmeters: (**a**) four tested UFMs; (**b**) five tested UFMs.

### 3.3. Numerical Simulation Verification by Experiments for Two to Three Tested DN 150 UFMs

Different from the large diameter of the UFM, the UFMs of DN150 adopt a more conservative experimental working condition, under which only one and two tandem arrangement is set up for verification experiment. For two UFMs in tandem arrangement, the relative measurement errors $E$ are plotted in Figure 7 as a function of flowrate $Q$. Similar to 4.1, both the experimental data (distributed symbols) and numerical results (continuous lines), as shown in Figure 7, expressed as an obvious negative power-law relationship. Also, we can note that the increase in $n$ can induce an overall shift in the $E$–$Q$ curve compared with DN250.

Compared to different diameter and inlet velocity, Re can be calculated using $Re = 4\rho Q / v\pi D$, where $D$ is the pipe or UFMs internal diameter, v is the kinematic viscosity of the fluid, and $Q$ is the inlet volume flow. Although the density and the kinematic viscosity of natural gas change at different times, they can be considered to be constant on the same bypass pipeline. Here, comparing to three tandem arrangement of DN250, the Re of DN150 comes to 1.67 times of that at the same inlet volume flow rate in the theory. Thus, in the previous overall shift theory, three tandem arrangement of DN150 are more likely to get terrible *E-Q* curve, which we can get proof from Figure 7. Simulation result shows that the overall shift offset of the *E–Q* curve is larger than DN250. This suggests that as the gas flows through smaller diameter UFMs, the flow distortion becomes more sensitive when *n* increases. However, it is still within the allowable range of relative error. Thus, three tandem arrangement of DN150 is numerically proved to be valid.

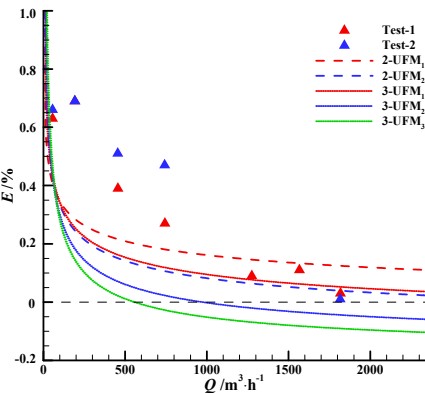

**Figure 7.** Relative measurement error *E* as a function of flowrate *Q* for two to three tested flowmeters.

## 4. Discussion

### 4.1. Definition and Distribution of Key Factors to Describe Variation in the Velocity Profile

For the UFMs under test, Daniel 3400, there are three important factors proposed by Zanker [18] and Daniel Inc. for examining the variation in the tested UFM performance with respect to the distortion in velocity profile. These three helpful ratios are defined as

$$K_S = (U_B + U_C)/(U_A + U_D), \tag{6}$$

$$K_A = (U_A + U_B)/(U_C + U_D), \tag{7}$$

$$K_C = (U_A + U_C)/(U_B + U_D), \tag{8}$$

where $K_S$, $K_A$, and $K_C$, are the swirl ratio, asymmetry ratio, and crossflow ratio, respectively. The swirl ratio compares the inner chords to the outer chords and is an indicator of swirl due to both the different radial locations and planes. In good conditions, $K_S$ should be close to $1.042/0.89 = 1.17$ [19]. The asymmetry ratio $K_A$ compares the flow in the top half of the pipe (path A and B) with that in the bottom half (path C and D) and, in good condition, should be close to 1. The crossflow ratio is the ratio of the flow velocity along the paths in one plane divided by those in the other plane at right angles; in good conditions, it should be close to 1. Here, the $K_S$ is shown in Figure 7 for both DN250 and DN150, which was reported to reveal the power-law velocity profile rule for fully developed turbulent flow by Moore et al. [20]. Furthermore, the swirl ratio is also presented as the "profile factor" by Daniel Inc., which is defined as the reciprocal of the meter factor, and which is sometimes called the velocity distribution correction factor in other documents [21]. The symbol shapes indicate the streamwise location of the tested UFMs (triangles: UFM1, diamonds: UFM2, circles: UFM3, squares: UFM4, gradient triangles: UFM5), and the symbol colors denote *n* (red: only one tested UFM, blue: two UFMs, green: three UFMs, purple: four UFMs, orange: five UFMs), as shown in Figure 8. We can see overall

decreasing trend of $K_S$ with increasing $Q$. However, these results are so rough that they cannot be used to evaluate the effects of chordal velocity profiles on $E$ (or $\delta_{\text{UFM}}$) under different $n$. In contrast, the $E$–$Q$ curves and simulated velocity profiles are more sensitive with respect to this theme.

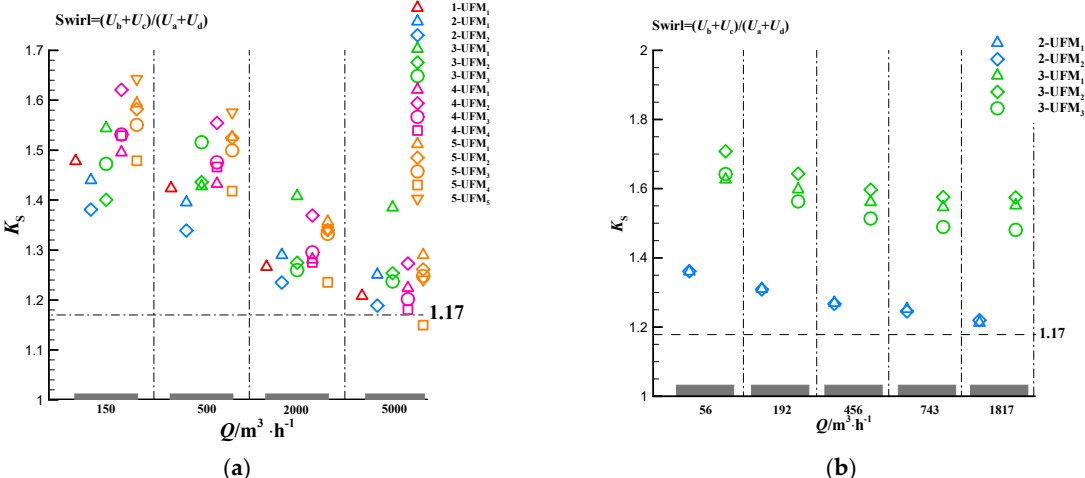

**Figure 8.** Factors associated with velocity profile as a function of Q: (**a**) swirl ratio for DN250; (**b**) swirl ratio for DN150.

### 4.2. Flow Analysis for Daniel 3400

In order to perform comparisons among velocity profiles along the same-type paths under different $Q$, the flow velocity $U$ as shown in Figure 9 was first nondimensionalized using the corresponding mean axial flow velocity $U_0$ and weighted by the corresponding w, and the location variable x was also nondimensionalized using the length of the corresponding path length L. As shown in equation 1, the values of the weight are 0.1382 along paths A and D ($w_A = w_D = 0.1382$) and 0.3618 along paths B and C ($w_B = w_C = 0.3618$), respectively. Note that the tested UFMs' structure is symmetrical at the origin, and each of the paths form a 60° with the axial flow; however, the velocity along the paths also approximately distributes according to Prandtl–von Karman logarithmic law, which is a common velocity distribution for fully developed turbulent pipe flow [14]. Despite this, there were pronounced differences between the velocity profiles along paths A and D (denoted as U-Profiles A and D) and that along paths B and C (denoted as U-Profiles B and C), because paths A and D were closer to the pipe wall with shorter L, and could be more sensitive to the boundary layer. To be more specific, U-Profiles A and D tend to be an inverted V shape, whereas their counterparts along paths B and C display an inverted $U$ shape. As we can see from Figure 9, when $n$ increases, the difference in U-Profiles of each path line also begins to increase, which is manifested as the overall deviation for U-Profiles A and D and the deviation of the main flow area for U-Profiles B and C. Further, we compare the U-Profiles A and B of three tested UFMs at the same flow rate (Figure 10). We find that the velocity profile of the main flow area hardly changed when $n = 3$, while the velocity profile of the near boundary layer area has obvious dispersion phenomenon. Obviously, the negative power-law $E$–$Q$ relationships can be partly derived from these results about the U-Profile at different $Q$ and the larger value of $w_B$ and $w_c$ (weighted coefficients).

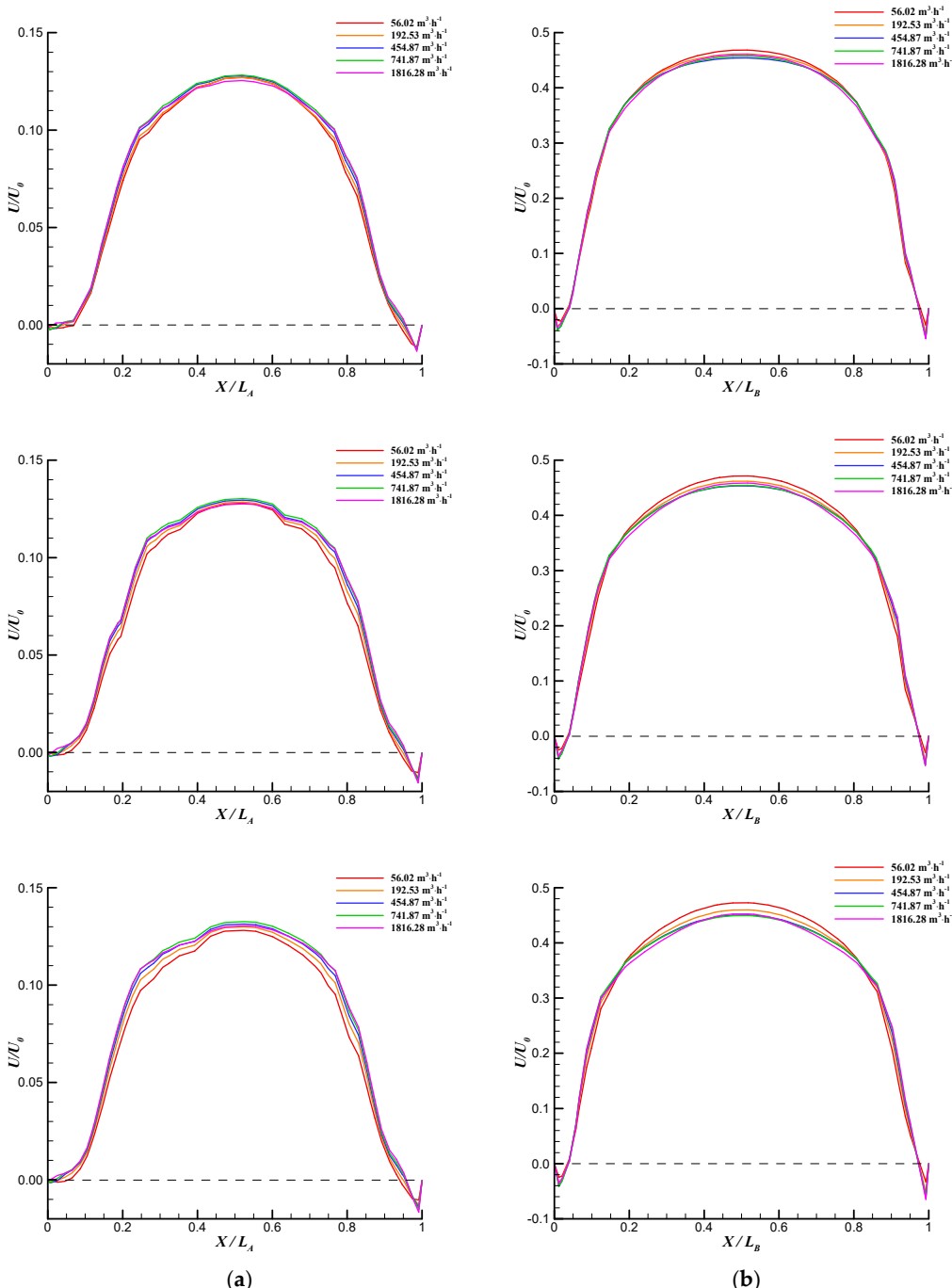

**Figure 9.** Velocity profiles along the same type of path under different *Q* for three tested UFMs (DN150): (**a**) A path; (**b**) B path.

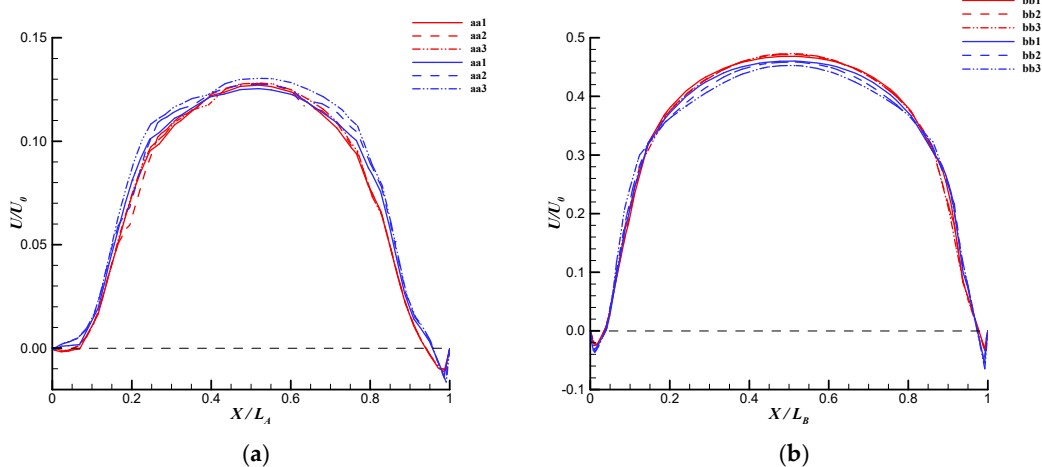

**Figure 10.** Velocity profiles along the same type of path for three tested UFMs (DN150) at different $Q = 56.02$ m$^3$/h for red line and $Q = 1816.28$ m$^3$/h for blue line: (**a**) A path; (**b**) B path.

As noted above, more tested UFMs would account for poorer performance of SVUFMST and we sought to explain this finding in accordance with the inlet flow condition of each UFM, especially the secondary flow, in the following discussion. In most practical engineering applications, the effect of the entry length on turbulent flow becomes insignificant beyond a pipe length of 10 times the diameter, as imposed by "JJG1030-2007." However, our detailed numerical results showed that the entry length would increase as Re increased, which was in line with the previous observations reported by Cimbala et al. [14], and hence we had ensured that all the entry lengths in our numerical simulations were larger than 50D. However, the entry velocity profiles of the UFMs at different streamwise locations can still be affected to some extent by the flow distortion caused by the transducer recesses, which can be treated as large-scale roughness elements. The mass-conservation law was satisfied by reducing the residual of the continuity equation by less than $10^{-6}$, and the relative deviation between the area-averaged inlet velocity at UFM1 and that at other UFMs was less than $10^{-3}$. The significant differences among the transverse velocity fields at the inlet of UFMs at different locations demonstrate occurrences of distinguishable secondary flows and remarkably diverse inlet flow conditions of each tested UFM, as shown in Figures 11 and 12. In particular, the phenomenon of secondary flows near the boundary is very obvious and it becomes more severe with the increase of the number of installations. Therefore, in spite of the low magnitude of the transverse velocity, it can be deduced that the inconsistency in the inlet flow condition, as a consequence of the secondary flows, may affect the slight differences (mostly below 1%) in $E$ among the tested UFMs while SVUFMST are executed.

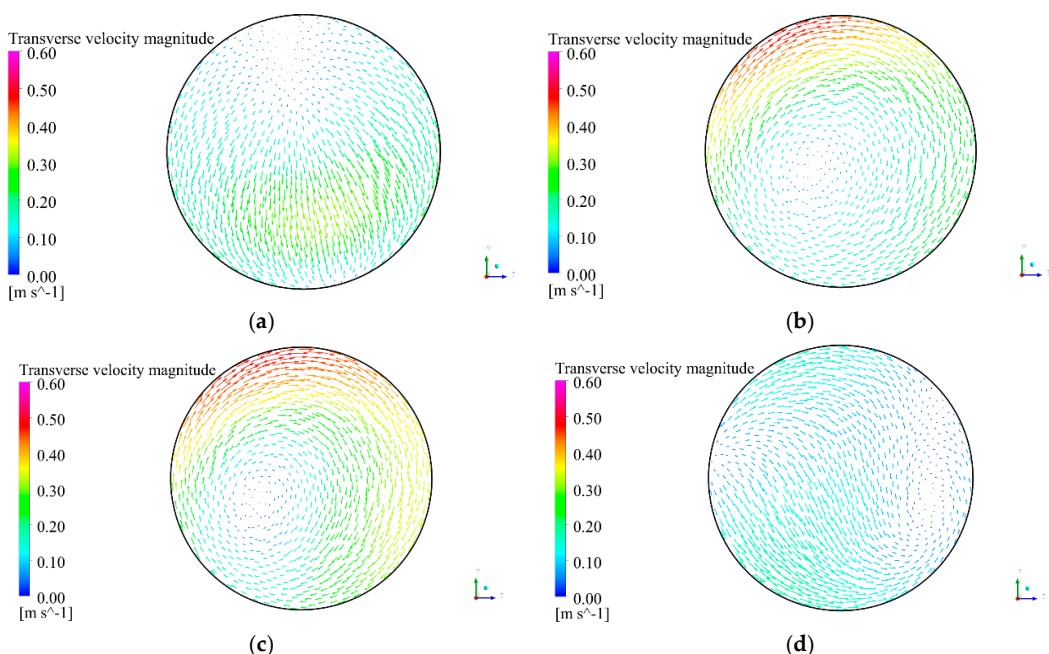

**Figure 11.** Transverse velocity distribution for four tested UFMs (DN250) at different UFM inlet with $Q$ = 5000 m$^3$/h: (**a**) UFM1; (**b**) UFM2; (**c**) UFM3; and (**d**) UFM4.

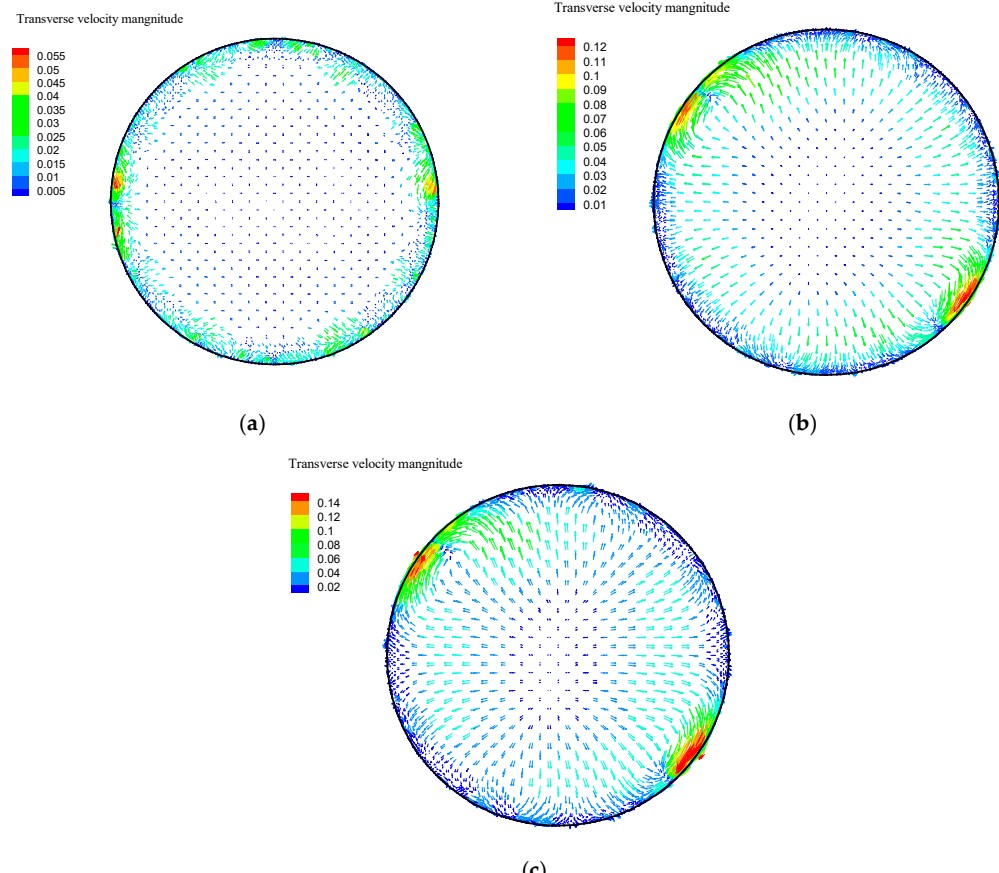

**Figure 12.** Transverse velocity distribution for four tested UFMs (DN150) at different UFM inlet with $Q$ = 1816.28 m$^3$/h: (**a**) UFM1; (**b**) UFM2; and (**c**) UFM3.

## 5. Conclusions

It is clear that SVUFMST has effectively solved the requirement of improving the efficiency of multi-UFMs' verification. As such, we recommend that the natural gas measurement station can use it in daily verification to reduce the disturbance of UFMs' verification on the natural gas supply. In this paper, a combined experimental and numerical analysis was performed to determine whether the flow field has a significant impact on the SVUFMST aiming at flowrate measurement of pressurized natural gas and, if so, to explain the mechanism of its action and the practicability of the SVUFMST. Compared with experimental methods, the simulation results using CFD can convincingly exclude the disturbance of industry manufactures and highly focuses on the aerodynamic factor. We pay particular attention to three test parameters, including the flowrate $Q$, the diameter $D$, and the number of the tested UFMs $n$, each of which significantly influence the characteristics of the internal flow fields and, in turn, the performance of SVUFMST.

Next, we will summarize each of the two models of tested UFMs. As for Daniel 3400(DN250), the actual flow experiments of $n \leq 3$ were carried out in NJMS, according to the method of SVUFMST. Data obtained from these actual-flow experiments were first used to verify the simulated results and then the verified numerical method was extended to simulate the internal flow of UFMs for $n = 4$ and $n = 5$, which would lead to excessive experimental cost. As for Daniel 3400(DN150), we compared the experimental and numerical simulation results of 2 tandem arrangement, and then extended the numerical method to $n = 3$. The velocity profiles along the paths and the $E$–$Q$ curves were extracted from the numerical results to reveal relationships between them and $n$ and $Q$. Comparison analysis shows that all the numerical results agree remarkably well with the available experimental data, and especially the overall $E$–$Q$ negative power-law distribution is in good accordance with the previous results [12,16,17]. For both UFMs, the increased in $Q$ can hardly change the shape characteristics of U-Profiles A and D, but that can significantly flatten U-Profiles B and C for any specified tested UFM. The negative power-law $E$–$Q$ relationships can be partly attributed to this result. It is of interest to note that more than two UFMs under test can induce an overall shift in an $E$–$Q$ curve and the two concentrated striped regions are consequently built on the abrupt shift in $E$–$Q$ curves associated with $n$, which also can be found for $n = 4$ and $n = 5$. However, different diameters of the tested UFMs can allow different values of n. A smaller diameter means a larger value of Re, which means a more violent turbulent environment at the same $Q$. Thus, DN150 can easier reach its tandem arrangement limit by more obvious secondary flows due to the accumulation of large-scale roughness elements. Our numerical simulation results show that the DN250 for $n \leq 4$ and the DN150 for $n \leq 3$ are proved to be appropriate in SVUFMST.

**Author Contributions:** Conceptualization, M.Y. and C.T.; methodology, T.H., Q.H., C.T., X.G., and Z.Y.; software, T.H. and X.G.; validation, T.H., C.T., and F.B.; formal analysis, C.T., F.B., T.H., and Q.H.; investigation, M.Y., Q.H. and T.H.; resources, M.Y.; data curation, T.H. and M.Y.; writing—original draft preparation, C.T. and T.H.

**Funding:** This research was supported by the National Natural Science Foundation of China (Grant No. 11602266, No. 11672284 and No. 11632016), S-T Projects of Petrochina West-East Natural Gas Pipeline Company (No. 201705) and Zhejiang Key Discipline of Instrument Science and technology.

**Acknowledgments:** We express our thanks to Nanjing natural gas measurement station (NJMS), which is affiliated with Petrochina West-East Natural Gas Pipeline Company, for providing us the WSF to perform the actual flow measurement and other necessary supports to carry out this research.

**Conflicts of Interest:** The authors declare no conflict of interest.

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
