# Peer review of "Simultaneous Verification of Multi-Ultrasonic-Flowmeters in Tandem Arrangement for Natural Gas"

_applsci, doi:10.3390/app9173632_

Round 1
Reviewer 1 Report
Please see attached comments.

Reviewer 2 Report
This work presents the practical viability of tandem arrangement for flowrate measurements of pressurized natural gas using multi-ultrasonic flowmeter. Results are highly fascinating and numerical results agree well with the experimental data. It can be accepted after minor revision. Reviewer’s major and minor comment are given below:
- Abstract- “It is important to note….” Restructure this sentence. It might be confusing for general readers.
- What would be the accuracy while measuring the multiphase liquids such as a slurry with entrained gas, or a fluid which tends to deposit.
- Further, it would be nice to present some data at low temperature (subsea temperature) to show the feasibility in subsea pipelines.
- Table 4: why coefficient of determination is higher than 1?
- Also weed out all the grammatical mistakes (or spelling errors) in the revised version, like “ comparef ” in conclusion section.
